# Factors associated with SARS-CoV-2 infection among people living with HIV: Data from the Balearic cohort (EVHIA)

Francisca Artigues Serra[1]*, Sophia Pinecki Socias[1], Francisco Javier Fanjul[1], Maria Peñaranda[1], Francisco Homar[2], Patricia Sorni[2], Julia Serra[3], Adelaida Rey[3], Lola Ventayol[4], Maria Dolores Macia[5], Maria Àngels Ribas[1], Melchor Riera[6]

1 Department of Internal Medicine, Section of Infectious Diseases, Son Espases University Hospital-IDISBA, Palma, Balearic Islands, Spain, 2 Department of Internal Medicine, Son Llatzer University Hospital-IDISBA, Palma, Balearic Islands, Spain, 3 Department of Internal Medicine, Inca Regional Hospital, Inca, Balearic Islands, Spain, 4 Department of Internal Medicine, Manacor Regional Hospital, Manacor, Balearic Islands, Spain, 5 Department of Microbiology, Son Espases University Hospital-IDISBA CIBERINFEC, Palma, Balearic Islands, Spain, 6 Department of Internal Medicine, Section of Infectious Diseases, Son Espases University Hospital-IDISBA CIBERINFEC, Palma, Balearic Islands, Spain

* francisca.artigues@ssib.es

**Data Availability Statement:** Data cannot be shared publicly because of legal considerations. According to the protocol approved by the Ethics

## Abstract

### Introduction

The impact of SARS-CoV-2 infection among people living with HIV (PLWH) has been a matter of research. We evaluated the incidence and factors associated with SARS-CoV-2 diagnosis among PLWH. We also assessed factors related to vaccination coverage in the Balearic Islands.

### Methods

A retrospective analytical study was performed, including patients from the Balearic cohort (EVHIA) who were visited at least twice between 1st January 2020 and 31st March 2022. Chi-square test and Mann-Whitney U test were used to compare categorical and continuous variables respectively. Multivariable Cox proportional hazards regression models were estimated to identify risk factors.

### Results

A total of 3567 patients with HIV were included. The median age was 51 years (IQR 44–59). Most of them were male (77,3%), from Europe (82,1%) or South America (13,8%). During the study period 1036 patients were diagnosed with SARS-CoV-2 infection (29%). The incidence rate was 153,24 cases per 1000 person-year. After multivariable analysis, men who have sex with men (MSM) were associated with an increased risk of SARS-CoV-2 infection (adjusted hazard ratio 1,324, 95% CI 1,138–1,540), whereas African origin, tobacco use and complete or booster vaccination coverage were negatively related. Overall, complete vaccination or booster coverage was recorded in 2845 (79,75%) patients. When analysing vaccination uptake, older patients (adjusted hazard ratio 5,122, 95% CI 3,170–8,288) and

Committee (IB 3808/18 PI) and the informed
consent signed by all participants, data transfer to
third parties is not allowed due to patient
confidentiality. Data are available from the Idisba
Data Access Committee (contact via email helemh.
vilchez@ssib.es) for researchers who meet the
criteria for access to confidential data.

**Funding:** The author(s) received no specific
funding for this work.

**Competing interests:** The authors have declared
that no competing interests exist.

those with a modified comorbidity index of 2–3 points (adjusted hazard ratio 1,492, 95% CI 1,056–2,107) had received more vaccine doses.

## Conclusions

In our study no HIV related factor was associated with an increased risk of SARS-CoV-2 infection, except for differences in the transmission route. Possible confounding variables such as mask wearing or social interactions could not be measured. Vaccines were of utmost importance to prevent SARS-CoV-2 infection. Efforts should be made to encourage vaccination in those groups of PLWH with less coverage.

## Introduction

The number of cases of SARS-CoV-2 infection increased worldwide, extensively affecting Spain. First laboratory-confirmed case of SARS-CoV-2 infection in Spain was reported on 31st January 2020 [1]. By the end of March 2022, a total of 6 epidemic periods had been described [2]. The Balearic Islands reported 268.000 SARS-CoV-2 confirmed cases and more than 1.300 deaths during these six waves [3].

A growing concern regarding the consequences of this novel infection among people living with HIV (PLWH) led to substantial research. Current evidence suggests that PLWH receiving effective antiretroviral therapy (ART) are not at higher risk of acquiring SARS-CoV-2 infection than general population [4]. However, socioeconomic and racial disparities have been described, with higher infection rates in low income areas and minority groups [5, 6]. Some risk factors associated with hospital admission and critical illness due to SARS-CoV-2, such as male sex and cardiovascular or respiratory diseases [6, 7], are prevalent among PLWH [8, 9]. Immunosuppression and detectable HIV viremia have been suggested as specific risk factors for severe outcomes [10, 11].

Since the beginning of the SARS-CoV-2 pandemic, the possibility that some antiretroviral drugs may have activity against SARS-CoV-2 infection has been studied, mainly including protease inhibitors and nucleotide/nucleoside reverse-transcriptase inhibitors [12, 13]. However, considering current evidence, no consistent recommendation about changing antiretroviral treatment can be made.

On the other hand, vaccines are a fundamental tool to reduce infection severity and mortality rates. COVID-19 vaccination in Spain started on 27th December 2020, including either 1 dose of Janssen (AD26.COV2.S) or 2 doses of Pfizer-BioNTech (BNT162b2), Moderna (mRNA-1273) or AstraZeneca (ChAdOx1). In a first stage, inmates, health and social health workers in elderly and disabled nursing homes were prioritized, as well as front-line healthcare workers and people with disabilities [14]. Vaccination was progressively opened according to age groups. A booster dose was introduced in October 2021, and an additional dose was advised for people belonging to very high-risk groups [15]. Overall, vaccination coverage in Spain has been high, with 92.9% of the population over 12 years fully vaccinated by 3rd January 2023 [16].

Despite increasing data on HIV and SARS-CoV-2 coinfection, some aspects are still poorly characterized. The aim of our study was to describe the incidence of SARS-CoV-2 in the Balearic HIV cohort (EVIHA) and define risk factors associated with SARS-CoV-2 primoinfection. Additionally, we analyzed epidemiological and clinical factors associated with vaccination uptake in our cohort.

## Material and methods

### Study population and design

We carried out a retrospective analytical study among PLWH included in the Balearic cohort. Almost all HIV care in the Balearic Islands is provided in public hospitals. The cohort has been monitored since 1998 through the eVIHa clinical platform, with the progressive participation of the four public hospitals in Majorca, and more recently from Minorca and Ibiza as well. It is an open, multicenter, observational cohort which prospectively includes all newly diagnosed patients aged 18 years or older, as well as transferred patients from other autonomous communities or other countries. It monitors sociodemographic data, comorbidities, cardiovascular risk factors, virological data, CD4 cell count, past and present antiretroviral treatments, among other data. Most information is automatically collected from electronic health care system records, laboratory and electronic prescription. All patients signed an informed consent document prior to their inclusion. They receive medical care at least every six months, with routine blood tests. Methodological, ethical and legal aspects of the eVHIa protocol code were approved by the Research Ethics Committee of the Balearic Islands.

Patients who had received telephone or in person medical visit at least twice between 1st January 2020 and 31st March 2022 were eligible for study inclusion. Patients were followed until SARS-CoV-2 infection diagnosis, death, transfer to another autonomous community or country, lost follow-up or until end of the study. This time points were chosen as the moment in which SARS-CoV-2 started to circulate in Europe, until the publication of a modification in the Spanish Healthcare System's protocol in which only severe cases or vulnerable groups should be tested. Individual notification in Spain was eliminated on 28th March 2022.

### Study variables

Information regarding SARS-CoV-2 infection was taken from the sanitary database of the Balearic autonomous community and from hospital reports. In most cases, confirmation of SARS-CoV-2 infection was obtained by means of a PCR test of respiratory samples, mainly nasopharyngeal specimens, and/or antigen detection, which was available in Spain in July 2021.

Sociodemographic variables included age, gender, origin, and tobacco use. Underlying medical conditions were also taken as variables. A modified Charlson index was calculated, considering acquired immunodeficiency syndrome (AIDS) as 1 point instead of 6 points. Among HIV-associated variables, we considered HIV transmission group, CDC category, naive CD4 cell count, tenofovir-based antiretroviral therapy at the beginning of the study, and HIV viral load (VL), CD4 cell count and CD4/CD8 ratio before SARS-CoV-2 infection or at the end of follow-up in those patients who didn't present coinfection. Undetectable viral load was considered when it was equal or lower than 50 copies per mL.

The type of vaccines before SARS-CoV-2 infection or until the end of follow-up, as well as the administration dates, were collected. The whole vaccination coverage regardless of SARS-CoV-2 infection was also recorded. Vaccination status was considered as unvaccinated or incomplete vaccination when receiving non doses or only 1 dose of Pfizer-Biontech (Cominarty), Moderna or AstraZeneca or others SARS-CoV-2 vaccines accepted in the European Union. Complete vaccination was considered when receiving 2 doses of the previous vaccines or 1 dose of Janssen. Individuals having received 3 or more doses of Pfizer-Biontech, Moderna or AstraZeneca or one dose of Janssen and a second dose of any other vaccine were referred as booster status.

Data was accessed between November 2022 and October 2023 for research purposes. Authors had also access to individual information if required.

## Statistical analysis

Categorical variables were expressed as total numbers (percentage), whereas continuous variables were expressed as median (interquartile range, IQR). Baseline characteristics of our cohort were described using proportions (accumulated incidence). Categorical variables were compared with Chi-square test, while Mann-Whitney U test was used in continuous variables. Hazard Ratios (HR) with 95% confidence interval (CI) were calculated to identify risk factors associated with SARS-CoV-2 diagnosis. Multivariable Cox proportional hazards regression models were estimated to identify risk factors, and discriminate cofounders. In the multivariable model, we adjusted for age, country of origin, HIV exposure group, tobacco use, CDC category, modified Charlson Index, plasma HIV viral load, tenofovir-based regimen, CD4 cell count and vaccination regimen.

We used univariate logistic regression to assess the factors associated with vaccination uptake. We calculated Odds ratios (OR) with 95% CI to assess the strength of association to the infection.

Records of missing values for adjustment covariates were excluded in the adjusted analyses, as there were few of them and they were not expected to affect estimates significantly. The level of significance of p was set at <0.05.

Statistical analysis was performed using SPSS 24.0.

## Results

### Descriptive analysis of our study population

Our study population included 3567 patients who met the inclusion criteria. A total of 3411 patients were still monitored by the end of the study, while 107 patients were lost follow-up and 49 died.

The main characteristics of our study population are shown in Table 1. Shortly, 2758 (77,3%) were male. The median age of the study population was 51 years (IQR 44–59). Overall, 2930 (82,1%) patients originated from Europe, 493 (13,8%) from South America, 117 (3,3%) from Africa and 27 (0,8%) from elsewhere. The main transmission route was through men who have sex with men (MSM) (41,6%), followed by heterosexual relations (31,4%) and injecting drug users (IDU) (20,9%). The median time living with HIV was 14,6 years (IQR 7,5–24,3). Regarding other underlying medical conditions, 1199 (33,6%) patients had a modified Charlson Index higher than 1, including hypertension, dyslipidemia, obesity, chronic HCV infection and chronic obstructive pulmonary disease as the most frequent comorbidities.

At the end of follow-up, the median CD4 cell count was 783 cells/uL (IQR 550–1061). Only 100 (2,82%) patients presented a count lower than 200 cells/uL. HIV-RNA in plasma was undetectable in 3328 (93,3%) patients. A total of 3518 (98,6%) patients were on ART, 2168 (60,77%) of them using a tenofovir-based regimen.

### SARS-CoV-2 infection risk factors

From 1st January 2020 until 31st March 2022, the incidence rate of SARS-CoV-2 infection among PLWH in our cohort was 153,24 cases per 1000 person-year, with a total number of 1036 (29%) affected patients. During 2020 the incidence was 105,62 cases per 1000 person-year, in 2021 135,21 cases per 1000 person-year, and in 2022 (until March) there were 469,60 cases per 1000 person-year (Table 2). Considering the six SARS-CoV-2 waves, 422 (40,7%) of our cases occurred during the time overlapping Omicron wave, which predominance started in early December 2021.

The cumulative incidence of SARS-CoV-2 primoinfection in our cohort was 28,3% in men, with no significance difference in women (p-value >0,05). The median age of SARS-CoV-2

**Table 1. Descriptive analysis of PLWH included in our study with and without SARS-CoV-2 infection.**

| | | Study population (N = 3567) | | SARS-CoV-2 negative (N = 2531) | | SARS-CoV-2 positive (N = 1036) | | p-value |
|---|---|---|---|---|---|---|---|---|
| **Sex** | | | | | | | | 0,064 |
| | Male | 2758 | 77,3% | 1978 | 71,7% | 780 | 28,3% | |
| | Female | 809 | 22,7% | 553 | 68,4% | 256 | 31,6% | |
| **Age, years** | | | | | | | | <0,005 |
| | **median (IQR)** | 51 | (44–59) | 52 | (45–59) | 49 | (41–57) | |
| | <35 | 336 | 9,4% | 196 | 58,3% | 140 | 41,7% | |
| | 36–50 | 1260 | 35,3% | 844 | 67,7% | 416 | 33,0% | |
| | 51–65 | 1666 | 46,7% | 1248 | 74,9% | 418 | 25,1% | |
| | 66–80 | 277 | 7,8% | 220 | 79,4% | 57 | 20,6% | |
| | 80–95 | 27 | 0,8% | 22 | 81,5% | 5 | 18,5% | |
| | missing | 1 | 0,0% | 1 | | | | |
| **Origin** | | | | | | | | <0,005 |
| | Europe | 2930 | 82,1% | 2095 | 71,5% | 835 | 28,5% | |
| | Africa | 117 | 3,3% | 95 | 81,2% | 22 | 18,8% | |
| | South America | 493 | 13,8% | 317 | 64,3% | 176 | 35,7% | |
| | Others | 27 | 0,8% | 24 | 88,9% | 3 | 11,1% | |
| **HIV transmission route** | | | | | | | | <0,001 |
| | HTX | 1121 | 31,4% | 816 | 72,8% | 305 | 27,2% | |
| | IDU | 747 | 20,9% | 562 | 75,2% | 185 | 24,8% | |
| | MSM | 1484 | 41,6% | 999 | 67,3% | 485 | 32,7% | |
| | others | 50 | 1,4% | 32 | 64,0% | 18 | 36,0% | |
| | unknown | 165 | 4,6% | 122 | 73,9% | 43 | 26,1% | |
| **Tobacco use** | | | | | | | | 0,515 |
| | Smoker | 1190 | 33,4% | 859 | 72,2% | 331 | 27,8% | |
| | Ex-smoker | 530 | 14,9% | 374 | 70,6% | 156 | 29,4% | |
| | Non-smoker | 1847 | 51,78 | 1298 | 70,3% | 549 | 29,7% | |
| **CDC clinical category** | | | | | | | | 0,001 |
| | Category A | 2259 | 63,3% | 1561 | 69,1% | 698 | 30,9% | |
| | Category B / C | 1308 | 36,7% | 970 | 74,2% | 338 | 25,8% | |
| **Years since HIV diagnosis** | | 14,6 | (7,5–24,3) | 15,3 | (8,3–24,9) | 13 | (6,4–22,4) | <0,005 |
| **CD4 nadir** | | 274 | (131–427) | 267 | (124–416) | 297 | (161–466) | <0,005 |
| **CD4 cell count before infection or end of study period (cells per uL)** | | | | | | | | 0,079 |
| | **median (IQR)** | 783 | (550–1061) | 777 | (540–1051) | 794 | (567–1078) | |
| | <200 | 100 | 2,82% | 71 | 71,0% | 29 | 29% | |
| | 200–349 | 240 | 6,76% | 181 | 75,4% | 59 | 24,6% | |
| | 350–500 | 384 | 10,82% | 290 | 75,5% | 94 | 24,5% | |
| | >500 | 2825 | 79,60% | 1985 | 70,3% | 840 | 29,7% | |
| **CD4/CD8 before infection or end of study period** | | | | | | | | 0,301 |
| | **median (IQR)** | 0,84 | (0,56–1,18) | 0,84 | (0,54–1,18) | 0,84 | (0,59–1,2) | |
| | <0.8 | 1528 | 42,8% | 1124 | 73,6% | 404 | 26,4% | |
| | 0.8–1 | 571 | 16,0% | 432 | 75,7% | 139 | 24,3% | |
| | >1 | 1238 | 34,7% | 894 | 72,2% | 344 | 27,8% | |
| | missig | 230 | 6,4% | 81 | 35,21% | 149 | 64,78% | |

(*Continued*)

**Table 1.** (Continued)

| | Study population (N = 3567) | | SARS-CoV-2 negative (N = 2531) | | SARS-CoV-2 positive (N = 1036) | | p-value |
|---|---|---|---|---|---|---|---|
| **VL>50 before infection or end of study period** | | | | | | | 0,981 |
| No | 3328 | 93,3% | 2371 | 71,2% | 957 | 28,8% | |
| Yes | 222 | 6,2% | 158 | 71,2% | 64 | 28,8% | |
| missing | 17 | 0,5% | 2 | 11,8% | 15 | 88,2% | |
| **VL detectable at any point during the study period** | | | | | | | 0,45 |
| No | 2899 | 81,3% | 2065 | 71,2% | 834 | 28,8% | |
| Yes | 668 | 18,7% | 466 | 69,8% | 202 | 30,2% | |
| **Tenofovir-based regimen** | | | | | | | 0,608 |
| No | 1350 | 37,8% | 963 | 71,3% | 387 | 28,7% | |
| yes | 2168 | 60,8% | 1529 | 70,5% | 639 | 29,5% | |
| missing | 49 | 1,4% | 39 | 79,6% | 10 | 20,4% | |
| **Vaccionation regimen before infection or end of study period** | | | | | | | 0,001 |
| Unvaccinated or incomplete | 1103 | 30,9% | 488 | 44,2% | 615 | 55,8% | |
| Complete | 1002 | 28,1% | 690 | 68,9% | 312 | 31,1% | |
| Booster | 1462 | 41,0% | 1353 | 92,5% | 109 | 7,5% | |
| **Modified Charlson Index** | | | | | | | <0,005 |
| 0–1 | 2368 | 66,4% | 1626 | 68,7% | 742 | 31,3% | |
| 2–3 | 354 | 9,9% | 261 | 73,7% | 93 | 26,3% | |
| >3 | 845 | 23,7% | 644 | 76,2% | 201 | 23,8% | |
| **Chronic comorbidities** | | | | | | | |
| Hypertension | 487 | 13,7% | 378 | 77,6% | 109 | 22,4% | <0,005 |
| Dyslipidaemia | 460 | 12,9% | 344 | 74,8% | 116 | 25,2% | 0,053 |
| Obesity | 566 | 15,9% | 399 | 70,5% | 167 | 29,5% | 0,792 |
| Chronic obstructive pulmonary disease | 384 | 10,8% | 271 | 70,6% | 113 | 29,4% | 0,861 |
| HCV infection | 695 | 19,5% | 520 | 74,8% | 175 | 25,2% | 0,012 |
| Chronic kidney disease | 223 | 6,3% | 176 | 78,9% | 47 | 21,1% | 0,007 |
| Chronic ischaemic heart disease | 60 | 1,7% | 45 | 75,0% | 15 | 25,0% | 0,486 |
| Diabetes | 321 | 9% | 203 | 71,7% | 91 | 28,3% | 0,774 |

HTX = heterosexual. IDU = injecting drug user. CDC clinical category for VIH symptoms. Catergory A = asymptomatic. Categroy B/C = Symptomatic. MSM = men who have sex with men. VL = viral load. P-value for categorical variables referes to chi-square test, and for continuous variables to the Mann-Whitney U test.

**Table 2. Incidence rates for primoinfection with SARS-CoV-2 during different period times.**

| Inicidence of Sars-CoV-2 priminfection | | | | | |
|---|---|---|---|---|---|
| Period | Cases | Time person-day | Inc. 1000 person-day | Inc. 1000 person-year | cumulative inc. (%) |
| **Total** | 1036 | 2.467.550,00 | 0,420 | 153,245 | 29,04 |
| **2020** | 335 | 1.157.644,00 | 0,289 | 105,624 | 9,85 |
| **2021** | 398 | 1.074.393,00 | 0,370 | 135,211 | 12,38 |
| **2022** | 303 | 235.510,00 | 1,287 | 469,598 | 12,125 |

Time person-day considers the days each individium was in the study during the period. Inc. = incidence

infected patients was 49 years (IQR 41–57), whereas the median age of non-infected patients was 52 years (IQR 45–59), being significantly different (p-value <0,005).

A comparison of the unadjusted and the adjusted multivariable analysis is shown in Table 3. The unadjusted analysis showed differences according to age, place of birth, HIV transmission route, tobacco use, CDC category, modified Charlson Index, detectable VL at any point during the study period and vaccination regimen. The comorbidities which were associated with a greater risk of infection were hypertension, dyslipidemia, HCV infection and chronic ischemic heart disease. However, in the adjusted multivariable analysis only MSM was associated with an increased risk of SARS-CoV-2 infection, while African origin, tobacco use and complete or booster vaccination regimen remained negatively associated with SARS-CoV-2 infection.

Eighteen (1,73%) patients required hospitalization because of COVID19 infection. Most of these patients had a mild-moderate infection, whereas only 2 of them required intensive care unit admission. Two out of 18 patients died because of SARS-CoV-2 infection.

## SARS-CoV-2 vaccination coverage

Taking all the study period into account, 722 (20,24%) patients received no vaccines or an incomplete regimen, whereas 2845 (79,75%) patients presented a complete vaccination status or received a booster. Differences regarding vaccination coverage are shown in Table 4. It is worth noticing that vaccination coverage was lower among female, people from Africa and South America, as well as in those patients with a previous SARS-CoV-2 diagnosis, although none of them remained statistically significant in the adjusted multivariable analysis. On the other hand, vaccination coverage was higher among older age groups, among those with a Charlson Index score of 2–3 points, in MSM and among tobacco users.

A total of 1929 (61,37%) patients received Pfizer-Biontech vaccine as their first dose, 543 (17,27%) patients received Moderna, 369 (12,19%) Astrazeneca, 295 (11,74%) Janssen and 7 received other than the previous stated. Booster vaccinations were predominantly done with Moderna (75,51%) or Pfizer (20,59%).

The incidence rate of SARS-CoV-2 infection was 103,24 cases per 1000 person-year in the period before an available vaccine. On the other hand, the incidence rate of SARS-CoV-2 infection was 131,73 cases per 1000 person-year in those without vaccine, 141,89 cases per 1.000 person-year with incomplete vaccination status, 116,69 cases per 1.000 person-year with complete vaccination status and 65,50 cases per 1000 person-year with booster status (Table 5).

## Discussion

Our study showed that SARS-CoV-2 infection was more frequent among MSM, while African origin, tobacco use and vaccination were negatively associated with SARS-CoV-2 diagnosis. When we analyzed COVID-19 vaccination coverage, older patients and those with more comorbidities had received more doses of vaccine, which probably explains why we found a negative association between these variables and SARS-CoV-2 infection in the unadjusted analysis but not in the adjusted multivariable analysis, highlighting the protective effect of vaccination.

The incidence of SARS-CoV-2 infection among PLWH has been variable across Europe, ranging from 0,3% to 5,7% person years [4]. However, most of this studies refer to the beginning of the pandemic, with little published information about the actual situation. Rial-Crestelo et. al. [17] reported an infection rate of 6,74%, with data from a tertiary hospital in Madrid (Spain) until February 2021. As far as we know, this is the first study to evaluate the COVID-

**Table 3. Factors associated with SARS-CoV-2 infection.**

| | SARS-CoV-2 diagnosis | | | | | |
|---|---|---|---|---|---|---|
| | HR (N = 3567) | (95% CI) | p-value | aHR (N = 3497) | (95% CI) | p-value |
| **Sex** | | | | | | |
| Male | ref | | | | | |
| Female | 1,134 | (0,984–1,305) | 0,082 | | | |
| **Age, years** | | | | | | |
| <35 | **ref** | | | ref | | |
| 36–50 | **0,645** | **(0,533–0,782)** | **0,0001** | 0,917 | (0,745–1,129) | 0,413 |
| 51–65 | **0,441** | **(0,364–0,534)** | **0,0001** | 0,847 | (0,675–1,062) | 0,151 |
| 66–80 | **0,354** | **(0,26–0,482)** | **0,0001** | 0,847 | (0,601–1,193) | 0,341 |
| 80–95 | **0,316** | **(0,129–0,77)** | **0,011** | 0,542 | (0,217–1,353) | 0,189 |
| **Place of birth** | | | | | | |
| Europe | ref | | | ref | | |
| Africa | 0,678 | (0,444–1,035) | 0,072 | **0,457** | **(0,292–0,714)** | **0,001** |
| South America | **1,536** | **(1,305–1,807)** | **0,0001** | 1,052 | (0,883–1,254) | 0,57 |
| Others | 0,39 | (0,126–1,212) | 0,104 | **0,296** | **(0,095–0,921)** | **0,036** |
| **HIV transmission route** | | | | | | |
| HTX | ref | | | ref | | |
| IDU | 0,851 | (0,709–1,022) | 0,084 | 0,923 | (0,759–1,124) | 0,425 |
| MSM | **1,282** | **(1,111–1,479)** | **0,001** | **1,324** | **(1,138–1,54)** | **<0,0005** |
| others | 1,355 | (0,842–2,179) | 0,211 | 1,026 | (0,606–1,737) | 0,925 |
| unknown | 0,958 | (0,696–1) | 0,792 | 0,979 | (0,7–1,37) | 0,902 |
| **Tobacco use** | | | | | | |
| Non-smoker | ref | | 0,052 | ref | | |
| Smoker | **0,848** | **(0,74–0,972)** | **0,018** | **0,802** | **(0,665–0,967)** | **0,021** |
| Ex-smoker | 0,984 | (0,749–1,068) | 0,218 | **0,747** | **(0,614–0,908)** | **0,003** |
| **Tobacco use** | | | | | | |
| Non-smoker | ref | | | | | |
| Ever-smoker | **0,863** | **(0,763–0,957)** | **0,018** | | | |
| **CDC clinical category** | | | | | | |
| Category A | ref | | | ref | | |
| Category B / C | **0,773** | **(0,679–0,88)** | **0,00001** | 1,024 | (0,853–1,229) | 0,798 |
| **Modified Charlson Index** | | | | | | |
| 0–1 | ref | | | ref | | |
| 2–3 | **0,778** | **(0,627–0,965)** | **0,023** | 1,124 | (0,884–1,429) | 0,341 |
| >3 | **0,685** | **(0,586–0,801)** | **0,0001** | 0,849 | (0,685–1,052) | 0,135 |
| **CD4/CD8 before infection or end of study period** | | | | | | |
| <0.8 | ref | | | | | |
| 0.8–1 | 0,886 | (0,731–1,075) | 0,22 | | | |
| >1 | 1,034 | (0,895–1,194) | 0,651 | | | |
| **VL>50 before infection or end of study period** | | | | | | |
| No | ref | | | ref | | |
| Yes | 1,108 | (0,86–1,427) | 0,427 | 0,799 | (0,588–1,086) | 0,152 |
| **VL detectable at any point during the study period** | | | | | | |

(*Continued*)

**Table 3.** (Continued)

| | | SARS-CoV-2 diagnosis | | | | | |
|---|---|---|---|---|---|---|---|
| | | HR (N = 3567) | (95% CI) | p-value | aHR (N = 3497) | (95% CI) | p-value |
| | No | ref | | | ref | | |
| | Yes | **1,251** | **(1,073–1,459)** | **0,004** | 1,152 | (0,956–1,387) | 0,136 |
| **Tenofovir-based regimen** | | | | | | | |
| | No | ref | | | ref | | |
| | Yes | 1,08 | (0,952–1,226) | 0,231 | 0,999 | (0,878–1,137) | 0,99 |
| **Vaccination regimen before infection or end of study period** | | | | | | | |
| | Unvaccinated or incomplete | ref | | | ref | | |
| | Complete | **0,379** | **(0,33–0,434)** | **0** | **0,359** | **(0,312–0,412)** | **<0,0005** |
| | Booster | **0,08** | **0,065–0,098** | **0** | **0,077** | **(0,062–0,095)** | **<0,0005** |
| **CD4 cell count (cells per uL)** | | | | | | | |
| | <200 | ref | | | | | |
| | 200–349 | 0,744 | (0,477–1,16) | 0,192 | | | |
| | 350–500 | 0,743 | (0,49–1,1279 | 0,162 | | | |
| | >500 | 0,899 | (0,621–1,301) | 0,572 | | | |
| **CD4 <200 cell** | | | | | | | |
| | Yes | ref | | | ref | | |
| | No | 0,871 | (0,602–1,26) | 0,472 | 0,793 | (0,532–1,181) | 0,253 |
| **Chronic comorbidities** | | | | | | | |
| | Hypertension | **0,028** | **(0,02–0,039)** | **0** | | | |
| | Dyslipidaemia | **0,788** | **(0,65–0,956)** | **0,016** | | | |
| | Obesity | 1,023 | (0,867–1,207) | 0,79 | | | |
| | Chronic obstructive pulmonary disease | 0,997 | (0,82–1,212) | 0,974 | | | |
| | HCV infection | **0,762** | **(0,648–0,896)** | **0,001** | | | |
| | Chronic kidney disease | **0,648** | **(0,483–0,868)** | **0,004** | | | |
| | Chronic ischaemic heart disease | 0,807 | (0,485–1,344) | 0,41 | | | |
| | Diabetes | 0,939 | (0,758–1,165) | 0,569 | | | |

HRs were calculated using Cox proportional hazards models. HR = hazard ratio. HTX = heterosexual. IDU = injecting drug user. CDC clinical category for VIH symptoms. Catergory A = asymptomatic. Categroy B/C = Symptomatic. MSM = men who have sex with men. VL = viral load. aHR = adjusted hazard ratio.

19 impact among PLWH during such a long time period. The incidence of SARS-CoV-2 infection in our cohort was 153 cases per 1000 person years (29,04%). Considering the omicron wave start (December 2021), 422 (40,7%) of the infections were diagnosed in this period. Another factor that increases as much the infection number could be the widespread use of antigen tests, which were more accessible and could let to more diagnosis.

According to the Ministry of Health of the Balearic Islands, a total of 268.310 SARS-CoV-2 confirmed cases were reported from January 2020 until March 2022 in the general population [3]. This represents an approximate rate of 101,6 cases per 1.000 person years, lower than the incidence reported among PLWH in our cohort. Different results have been described in other cohorts. Fernández-Fuentes et. al. [18] showed a lower seroincidence of SARS-CoV-2 infection among PLWH from May to November 2020 in Seville, pointing out that a possible explanation could be a higher compliance of preventing measures. Other reports also performed mainly during the first year of the pandemic found similar results, with lower SARS-CoV-2 incidence

**Table 4. Factors associated with SARS-CoV-2 vaccine coverage among PLWH.**

| | Total | No vaccine/ Incomplete Status (N = 722) | | Complete/ Booster Status (N = 2845) | | p-value | OR | 95% CI | p -value | aOR | 95% CI | p -value |
|---|---|---|---|---|---|---|---|---|---|---|---|---|
| **Birth place** | | | | | | | | | | | | |
| Europe | 2930 | 551 | 18,8% | 2379 | 81,2% | <0,0005 | ref | | | ref | | |
| Africa | 117 | 38 | 32,5% | 79 | 67,5% | | **0,482** | **(0,323–0,717)** | **<0,0005** | 0,685 | 0,447–1,05 | 0,685 |
| South-America | 493 | 126 | 25,6% | 367 | 74,4% | | **0,675** | **(0,54–0,843)** | **0,001** | 0,896 | 0,703–1,141 | 0,372 |
| Other | 26 | 7 | 26,9% | 19 | 73,1% | | 0,692 | (0,263–1,503) | 0,297 | 0,883 | 0,359–2,17 | 0,786 |
| **Sex** | | | | | | | | | | | | |
| Female | 809 | 196 | 24,2% | 613 | 75,8% | **0,001** | **0,737** | **(0,611–0,888)** | **0,001** | 0,844 | 0,675–10,54 | 0,135 |
| Male | 2758 | 526 | 19,1% | 2232 | 80,9% | | ref | | | ref | | |
| **Age (15 year intervals) (missing 1)** | | | | | | | | | | | | |
| <35 | 336 | 122 | 36,3% | 214 | 63,7% | <0,0005 | ref | | | ref | | |
| 36–50 | 1260 | 286 | 22,7% | 974 | 77,3% | | 0,739 | (0,314–1,737) | 0,487 | **2,088** | **1,587–2,749** | **<0,0005** |
| 51–65 | 1666 | 278 | 16,7% | 1388 | 83,3% | | 1,434 | (0,621–3,310) | 0,398 | **3,062** | **2,276–4,120** | **<0,0005** |
| 66–80 | 277 | 28 | 10,1% | 249 | 89,9% | | 2,102 | (0,911–4,85) | 0,082 | **5,122** | **3,17–8,288** | **<0,0005** |
| 81–95 | 27 | 8 | 29,6% | 19 | 70,4% | | **3,744** | **(1,52–9,337)** | **0,04** | 1,346 | 0,555–3,261 | 0,511 |
| **HIV transmission route** | | | | | | | | | | | | |
| HTX | 1121 | 257 | 22,9% | 864 | 77,1% | 0,064 | ref | | | ref | | |
| IDU | 747 | 141 | 18,9% | 606 | 81,1% | | **1,278** | **(1,016–1,609)** | **0,036** | 0,922 | 0,714–1,191 | 0,533 |
| MSM | 1484 | 279 | 18,8% | 1205 | 81,2% | | **1,285** | **(1,062–1,555)** | **0,01** | **1,381** | **1,093–1,745** | **0,007** |
| others | 50 | 13 | 26,0% | 37 | 74,0% | | 0,847 | (0,442–1,617) | 0,614 | 1,622 | 0,816–3,226 | 0,168 |
| unknown | 165 | 32 | 19,4% | 133 | 80,6% | | 1,236 | (0,82–1,863) | 0,311 | 1,157 | 00,756–1,771 | 0,501 |
| **Tobacco Use** | | | | | | | | | | | | |
| Non- Smoker | 1847 | 424 | 23,0% | 1423 | 77,0% | <0,0005 | ref | | | ref | | |
| Ever smoker | 1720 | 298 | 17,3% | 1422 | 82,7% | | **1,422** | **(1,205–1,678)** | **<0,0005** | **1,235** | **1,035–1,475** | **0,02** |
| **HIV symptomatic** | | | | | | | | | | | | |
| Asymptomatic (category A) | 2259 | 479 | 21,2% | 1780 | 78,8% | 0,06 | ref | | | ref | | |
| Sympotmatic (category B/C) | 1308 | 243 | 18,6% | 1065 | 81,4% | | 1,179 | (0,993–1,401) | 0,06 | 0,999 | 0,784–1,273 | 0,999 |
| **Tenofovir based treatment** | | | | | | | | | | | | |
| No | 1350 | 274 | 20,3% | 1076 | 79,7% | 0,999 | ref | | | ref | | |
| Yes | 2168 | 440 | 20,3% | 1728 | 79,7% | | 1 | (0,845–1,184) | 0,999 | | | |
| **Chronic comorbidities** | | (referenced to no comorbility condition) | | | | | | | | | | |
| Obesity | 566 | 89 | 15,7% | 477 | 84,3% | **0,004** | **1,433** | **(1,124–1,826)** | **0,004** | | | |
| Hypertension | 487 | 63 | 12,9% | 424 | 87,1% | **<0,0005** | **1,832** | **(1,287–2,42)** | **<0,0005** | | | |
| Dyslipidaemia | 460 | 55 | 12,0% | 405 | 88,0% | **<0,0005** | **2,013** | **(1,5–2,702)** | **<0,0005** | | | |
| Chronic ischaemic heart disease | 60 | 6 | 10,0% | 54 | 90,0% | **0,046** | **2,309** | **(0,989–5,388)** | **0,053** | | | |
| Chronic obstructive pulmonary disease | 384 | 66 | 17,2% | 318 | 82,8% | 0,115 | 1,251 | (0,947–1,653) | 0,116 | | | |
| Chronic liver disease | 33 | 9 | 27,3% | 24 | 72,7% | 0,312 | 0,674 | (0,312–1,456) | 0,316 | | | |
| Diabetes | 321 | 44 | 13,7% | 277 | 86,3% | **0,002** | **1,662** | **(1,196–2,31)** | **0,002** | | | |
| Chronic kidney disease | 223 | 28 | 12,6% | 195 | 87,4% | **0,003** | **1,824** | **(1,217–2,734)** | **0,004** | | | |
| HCV infection | 695 | 126 | 18,1% | 569 | 81,9% | 0,123 | 1,183 | (0,956–1,463) | 0,123 | | | |
| HBV infection | 85 | 16 | 18,8% | 69 | 81,2% | 0,742 | 1,097 | (0,633–1,901) | 0,742 | | | |
| Neuropsychiatric disease | 204 | 33 | 16,2% | 171 | 83,8% | 0,137 | 1,335 | (0,911–1,956) | 0,138 | | | |
| **Charlson Index** | | | | | | | | | | | | |

(*Continued*)

**Table 4.** (Continued)

| | Total | No vaccine/ Incomplete Status (N = 722) | | Complete/ Booster Status (N = 2845) | | p-value | OR | 95% CI | p -value | aOR | 95% CI | p -value |
|---|---|---|---|---|---|---|---|---|---|---|---|---|
| 0–1 | 2368 | 518 | 21,9% | 1850 | 78,1% | <0,0005 | ref | | | ref | | |
| 2–3 | 354 | 46 | 13,0% | 308 | 87,0% | | **1,875** | **(1,355–2,594)** | **<0,0005** | **1,492** | **1,056–2,107** | **0,023** |
| >3 | 845 | 158 | 18,7% | 687 | 81,3% | | 1,217 | (0,998–1,485) | 0,052 | 0,994 | 0,754–1,309 | 0,964 |
| **Prior Sars-CoV-2 infection** | | | | | | | | | | | | |
| No | 3022 | 525 | 17,4% | 2497 | 82,6% | <0,0005 | ref | | | | | |
| Yes | 545 | 197 | 36,1% | 348 | 63,9% | | **0,371** | **(0,305–0,453)** | **<0,0005** | | | |

among PLWH [17, 19, 20]. On the other hand, Nomah et. al. [21] reported less SARS-CoV-2 testing among PLWH from March to December 2020 in Catalonia but with a higher test positivity compared with general HIV-negative population.

Our study includes the introduction of SARS-CoV-2 vaccination in December 2021. By March 2022, 86,2% of the general population had complete status of vaccination in Majorca [3], whereas only 79,75% of our cohort had achieved it. There are several hypothetical explanations that account for this difference. Firstly, it could be attributed to a certain reluctance to SARS-CoV-2 vaccination among PLWH. Socioeconomic differences and education level may have also influenced. In addition, PLWH were not considered a priority group for the Spanish authorities during the immunization campaign until October 2021, when only PLWH with CD4 count less than 200/uL were included [15].

Some studies have analyzed factors associated with SARS-CoV-2 vaccination coverage among PLWH. Similar to Nomah et. al. [22], we found that being from outside of Europe and having a previous SARS-CoV-2 infection were associated with a lower vaccination coverage, although only in the unadjusted analysis; whereas the vaccine uptake increased with increasing age groups and increasing number of comorbidities. Nomah et. al. [22] also showed that CD4 cell count of 200-349/uL or 350-499/uL and detectable plasma HIV-RNA were associated with less vaccination. Contrary to us, Lv et. al. [23] described lower vaccination rates related to the presence of chronic disease and also to CD4 T cell count <200/uL. Jaiswal et. al. [24] also showed that vaccine uptake was associated with older age, higher education level and undetectable viral load.

It is worth noticing that a reduced risk of SARS-CoV-2 infection was observed in our study among those patients with any dose of SARS-CoV-2 vaccine, particularly in those with complete or booster vaccination. Increasing data regarding immunogenicity and safety of

**Table 5. Incidence rate of SARS-CoV-2 in relation with the different vaccine status covered.**

| SARS-CoV-2 primoinfection incidence | | | | |
|---|---|---|---|---|
| Status | Cases | Time (person-day) | Incidence (1000 person-day) | Incidence (1000 person-year) |
| Infection prior available vaccines | 324 | 1144672 | 0,283 | 103,249 |
| Without vaccine | 222 | 615101 | 0,361 | 131,734 |
| Incomplete Status | 291 | 748569 | 0,389 | 141,891 |
| Complete Status | 303 | 947710 | 0,320 | 116,697 |
| Booster status | 118 | 657530 | 0,179 | 65,503 |

Time was considered from the moment vaccines were available until the moment of infection for every person.

SARS-CoV-2 vaccines among PLWH is being published. Initially, it was hypothesized that the humoral response could be reduced as a result of a dysfunctional immune system. However, several studies have shown that SARS-CoV-2 vaccines among PLWH are acceptable, with a safe profile and an optimal immune response, especially in those cases on established ART, suppressed HIV viral load and high baseline CD4 counts [25, 26]. Fowokan et. al. [27] described that PLWH reached the vaccine efficacy peak later than healthy controls, with a faster waning degree over time. The third dose (booster shot) has also proven to increase antibody response among PLWH [28, 29], being of upmost importance.

In our study, co-infected patients were younger. Older age has been widely described as a risk factor for severe disease [6, 30–32]. Theodore et. al. [33] reported that age >65 years was associated with a decreased rate of co-infection, suggesting stricter adherence to regulations, which is probably what happened in our cohort, with higher vaccination coverage in older age groups. Conversely, another study described older age and opportunistic infections as risk factors for coinfection [19].

We observed that being from South America or MSM were possible risk factors for SARS-CoV-2 diagnosis. These results had been also described in previous reports [10, 34]. Nevertheless, only MSM remained significant in our multivariate analyses, despite higher vaccine coverage. Other variables such as social interactions, safety distance, mask wearing, hand hygiene or proper ventilation of the rooms had also a great impact during all the pandemic. Although they could not be measured in our study, these variables might have influenced on the above results, acting as confounders. On the other hand, African origin was associated with a reduced risk of SARS-CoV-2 infection, despite presenting a low vaccination coverage in the unadjusted analysis. This result has also been described in previous reports [33]. Other studies with a higher percentage of non-Hispanic black people found opposite results [35], probably associated with socioeconomic inequities in that area. More data is needed in order to study the eventual role of ethnic groups.

There are diverging results in the literature regarding the influence of comorbidities on SARS-CoV-2 diagnosis. Some authors showed that having 4 or more comorbidities was associated with an increased risk of infection [10]. Similar to us, others studies found no significant difference [18, 34]. In our case, this could be explained by a higher vaccination coverage among people with chronic diseases, probably because of a self-perception of risk.

Tobacco use showed unexpected results in our study, with smokers and ex-smokers being negatively associated with SARS-CoV-2 infection. It is generally known that cigarette smoking is a risk factor for respiratory infectious diseases. Therefore, one possible explanation to our results could be a higher vaccination uptake among these patients because of a sense of vulnerability. However, Fernández-Fuentes et. al. published similar results during de pre-vaccination period [18]. Despite possible biases, several hypothesis to explain both the protective and the harmful effect of tobacco use on SARS-CoV-2 infection have been published [36]. In any case, further investigation is still needed.

Shortly after the beginning of the pandemic, tenofovir was proposed as a promising treatment for COVID-19 [37, 38]. Some studies have shown lower severity of SARS-CoV-2 infection in PLWH treated with tenofovir [39–41]. Lea et. al. [41] described that the protective effect of tenofovir alafenamide (TAF) and tenofovir disoproxil fumarate (TDF) were similar in magnitude. However, other reports didn't show a clinical benefit after adjusting for specific comorbidities, mainly chronic kidney disease [42, 43]. The PANCOVID study also showed no evidence that treatment with TDF and emtricitabine (FTC) improves outcomes among hospitalized patients [44]. In our study, tenofovir-based ART had no influence on the risk of SARS-CoV-2 infection. CD4 count or HIV viral load didn't show any association with SARS-CoV-2 infection either, in line with the previously mentioned studies [10, 18].

This study has some limitations. First of all, the first case of SARS-CoV-2 infection in the Balearic Islands was reported on 7[th] February 2020. Therefore, we have probably overestimated the period of time at risk of SARS-CoV-2 infection and the period of time without vaccines. Second, as this is a retrospective study, data was not collected for analyses proposes. In line with this, hospitalized patients are probably underestimated because secondary diagnosis such as HIV were not fully registered at that time. In addition, 322 people entered our cohort after 1[st] January 2022. Some of them were new HIV diagnoses, but others were PLWH already diagnosed who came from other autonomous communities or from other countries. In these cases, there was a lack of information from the previous study period which could let to an underdiagnoses of SARS-CoV-2 infection. Third, incidence and vaccination coverage comparisons between general population and PLWH included in the EVHIA were not standardized. Finally, most of the PLWH included in this study were on effective ART and, therefore, results can't be generalized.

## Conclusions

To sum up, no HIV related factor was associated with an increased risk of SARS-CoV-2 infection in our study, except for differences in the transmission route. Possible confounding variables such as mask wearing or social interactions could not be measured. Conversely, African origin and vaccination were associated with a reduced risk of infection, highlighting the importance of immunization in the control of SARS-CoV-2 infection. Vaccination coverage in our cohort was higher among older people and those with a higher number of comorbidities, who were considered priority groups. However, more emphasis should be placed on those groups with less vaccination coverage.

## Acknowledgments

The authors thank the study participants and their families; site staff who provide healthcare assistance; and programmer analysts who constantly update the EVIHA platform. They also have a special acknowledgement for Aina Millan, from the Clinical Research and Clinical Trials Unit (UICEC), Idisba.

## Author Contributions

**Conceptualization:** Melchor Riera.

**Data curation:** Francisca Artigues Serra, Sophia Pinecki Socias, Francisco Javier Fanjul, Maria Peñaranda, Francisco Homar, Patricia Sorni, Julia Serra, Adelaida Rey, Lola Ventayol, Maria Dolores Macia, Maria Àngels Ribas, Melchor Riera.

**Formal analysis:** Sophia Pinecki Socias.

**Investigation:** Francisca Artigues Serra.

**Methodology:** Francisco Javier Fanjul, Melchor Riera.

**Supervision:** Francisco Javier Fanjul, Melchor Riera.

**Writing – original draft:** Francisca Artigues Serra, Sophia Pinecki Socias.

**Writing – review & editing:** Francisco Javier Fanjul, Maria Peñaranda, Francisco Homar, Patricia Sorni, Julia Serra, Adelaida Rey, Lola Ventayol, Maria Dolores Macia, Maria Àngels Ribas, Melchor Riera.

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
