## [Decision Letter · Decision Letter 0]

29 Apr 2024

PONE-D-23-43118Factors associated with SARS-CoV-2 infection among people living with HIV: Data from the Balearic cohort (EVHIA)

PLOS ONE

Dear Dr. Artigues Serra,

Thank you for submitting your manuscript to PLOS ONE. After careful consideration, we feel that it has merit but does not fully meet PLOS ONE’s publication criteria as it currently stands. Therefore, we invite you to submit a revised version of the manuscript that addresses the points raised during the review process.

**Please revise according to the feedback provided. Advise focus on more completely detailing the methods applied to the manuscript. **

We look forward to receiving your revised manuscript.

Kind regards,

Jake Michael Pry, PhD, MPH

Academic Editor

PLOS ONE

Journal Requirements:

Reviewers' comments:

Reviewer's Responses to Questions

**Comments to the Author**

1. Is the manuscript technically sound, and do the data support the conclusions?

Reviewer #1: Partly

Reviewer #2: Yes

Reviewer #3: Yes

2. Has the statistical analysis been performed appropriately and rigorously? 

Reviewer #1: Yes

Reviewer #2: Yes

Reviewer #3: Yes

3. Have the authors made all data underlying the findings in their manuscript fully available?

Reviewer #1: Yes

Reviewer #2: Yes

Reviewer #3: Yes

4. Is the manuscript presented in an intelligible fashion and written in standard English?

Reviewer #1: Yes

Reviewer #2: Yes

Reviewer #3: Yes

5. Review Comments to the Author

**Reviewer #1: **1- Line 44-47: “Older people and those with more comorbidities had higher vaccination uptake after the multivariable analysis; whereas females, African and South American people, and prior SARS-CoV-2 infection were associated with less vaccination coverage only in the unadjusted analysis”. These are not the main results in the present study. The authors should focus on to explore the risk factors (such as age, comorbidities, vaccination, etc.) modulating the infection of SARS-CoV-2 by multivariable analysis.

2- Line 57-58: “The Balearic Islands reported 268.000 SARS-CoV-2 confirmed cases and more than 1.300 deaths during these six waves”. The authors should check whether the number of SARS-CoV-2 confirmed cases and deaths was correct.

3- In Table 3, would the authors please check whether aHZR is right.

4- Other variables (such as social interactions, mask wearing, etc.) should be considered when the authors explored the risk factors associated with SARS-CoV-2 infection among people living with HIV.

5- Would the authors please discuss the point of MSM being a risk factor for SARS-CoV-2 infection in the section of discussion. In Table 3, the authors observed MSM was an independent risk factor for SARS-CoV-2 infection, but in Table 4, the rate of vaccination coverage was the highest among HIV transmission routes. The authors should address this.

6- There are some typographical or grammatical errors in the present manuscript, would the authors please correct.

**Reviewer #2:** 1. The method section stated that the paper is a descriptive retrospective study, however, I think your study is analytic.

2. In Table 3 aHZR should be replaced by aHR.

3. in case of citing more than one reference in the text, please insert all citations in Parentheses for example (5, 6) no (5)(6).

4. In Table 1, the authors reported column percentages, however, I think row percentages can be more informative.

5. The authors covered two research questions of COVID-19 incidence and vaccine coverage among PLHIV. I think the authors presented a lot of information in the manuscript; I suggest you present these two research questions in different manuscripts.

**Reviewer #3**: 1. This study describes a large cohort of HIV+ patients over 2 years to determine incidence and risk factors for sars cov2 infection. Data taken from an established data base with informed consent. 96% patients remained in the cohort at the end of the study period (3411).

The questions set out are investigated and answered well for the purpose of this manuscript .

A few comments:

3. Table 1 : what is p value referring to? eg age; origin and transmission

4. interesting that COPD is not a risk factor is there any data re Lung function in this cohort?

5. Are there any data re patients using HCV drugs?

4.There are other arv therapies which are thought to have sars-cov 2 activity. eg Abacavir and zidovudine and Protease inhibitors. Maybe include a comment or reference to this. Especially for those patients who were on 2nd / 3rd line regimens , and for patients with vl>50

6. PLOS authors have the option to publish the peer review history of their article (what does this mean?). If published, this will include your full peer review and any attached files.

Reviewer #1: No

Reviewer #2: No

Reviewer #3: No

---

## [Author Response · Author response to Decision Letter 0]

12 Jun 2024

Dear Editor and Reviewers,

Thank you very much for taking the time to review our manuscript and for providing constructive feedback. It is a pleasure for us to have the opportunity to improve it. We have carefully read your valuable suggestions and comments, and we have thoroughly revised the manuscript accordingly. Please see below, in blue, our detailed responses and concerns.

ACADEMIC EDITOR’S COMMENTS

There is controversy regarding the effect of COVID-19 infection on people living with HIV, so the impact of SARS-CoV-2 infection among people living with HIV (PLWH) has been a matter of research. In this study the authors calculated COVID-19 incidence and related factors among PLWH, also they evaluated vaccine coverage and related factors among PLWH. The results showed that the incidence of COVID-19 among PLWH was 153 cases per 1000 person-years which was more than the general population. Also, they showed that Men who have sex with men (MSM) were associated with an increased risk of SARS-CoV-2 infection, whereas African origin, tobacco use, and complete or booster vaccination coverage were negatively related to SARS-CoV-2 infection among PLWH. I think this is a valuable study, however, I have some minor comments:

1. The method section stated that the paper is a descriptive retrospective study, however, I think your study is analytic: Suggestion has been accepted and changed. 

2. In Table 3 aHZR should be replaced by aHR. : Correction done.

3. In case of citing more than one reference in the text, please insert all citations in Parentheses for example (5, 6) no (5)(6). Correction done.

4. In Table 1, the authors reported column percentages, however, I think row percentages can be more informative: Thank you for the suggestion. We also consider it is more informative and have therefore adapted the suggestion.

5. The authors covered two research questions of COVID-19 incidence and vaccine coverage among PLHIV. I think the authors presented a lot of information in the manuscript; I suggest you present these two research questions in different manuscripts. We really appreciate your suggestion. We decided to present the two questions in the same manuscript because both issues are related, since vaccination coverage had a great influence on the decrease of COVID-19 cases. Therefore, we found it interesting to analyze which factors were associated with vaccination. We completely agree that there is a lot of information on display. Therefore, changes have been made to focus on these two main questions and to avoid too much additional information.

REVIEWERS' COMMENTS

Reviewer #1: 

1- Line 44-47: “Older people and those with more comorbidities had higher vaccination uptake after the multivariable analysis; whereas females, African and South American people, and prior SARS-CoV-2 infection were associated with less vaccination coverage only in the unadjusted analysis”. These are not the main results in the present study. The authors should focus on to explore the risk factors (such as age, comorbidities, vaccination, etc.) modulating the infection of SARS-CoV-2 by multivariable analysis. We are grateful for this comment as it has helped us to summarize better our findings. We had two main objectives: 1) to evaluate the incidence and factors associated with SARS-CoV-2 diagnosis among PLWH, and 2) to assess factors related to vaccination coverage.

Vaccination had a great impact reducing SARS-CoV-2 infection. That is why we thought that analyzing factors associated with vaccination in the same article could be interesting. Appropriate corrections have been made. Unnecessary information has been removed.

2- Line 57-58: “The Balearic Islands reported 268.000 SARS-CoV-2 confirmed cases and more than 1.300 deaths during these six waves”. The authors should check whether the number of SARS-CoV-2 confirmed cases and deaths was correct. This information is correct, the reference is from the official webpage from the Balearic Islands Government. Our Autonomous Community had fewer cases of SARS-CoV-2 infection than other areas of Spain. This is probably why the mortality rate was not so high.

3- In Table 3, would the authors please check whether aHZR is right. Thank you for the comment, the second column refers to the adjusted Hazard Ratio. Z has been deleted from the acronym.

4- Other variables (such as social interactions, mask wearing, etc.) should be considered when the authors explored the risk factors associated with SARS-CoV-2 infection among people living with HIV. We absolutely agree with the reviewer. Variables such as social interactions, safety distance, mask wearing, hand hygiene or proper ventilation of the rooms had a great impact during all the pandemic. Unfortunately, we have no way to measure them. We have added a specific mention about this issue in the new manuscript.

5- Would the authors please discuss the point of MSM being a risk factor for SARS-CoV-2 infection in the section of discussion? In Table 3, the authors observed MSM was an independent risk factor for SARS-CoV-2 infection, but in Table 4, the rate of vaccination coverage was the highest among HIV transmission routes. The authors should address this. Thank you very much for your comment. This point is probably related with the previous one. Social interactions or mask wearing could not be measured in our study and, in this case, they probably acted as confounding variables that generated an association between MSM and SARS-CoV-2 infection. We have clarified this issue in the new manuscript.

6- There are some typographical or grammatical errors in the present manuscript, would the authors please correct. We regret any careless errors in the manuscript. We have already tried to correct them all.

Reviewer #2: 

1. The method section stated that the paper is a descriptive retrospective study, however, I think your study is analytic. Suggestion has been accepted and changed.

2. In Table 3 aHZR should be replaced by aHR. Thank you for the comment, the second column referes to the adjusted Hazard Ratio. Z has been deleted from the acronym. 

3. in case of citing more than one reference in the text, please insert all citations in Parentheses for example (5, 6) no (5)(6). Correction done.

4. In Table 1, the authors reported column percentages, however, I think row percentages can be more informative. Thank you for the suggestion. We also consider it is more informative and have therefore adapted the suggestion.

5. The authors covered two research questions of COVID-19 incidence and vaccine coverage among PLHIV. I think the authors presented a lot of information in the manuscript; I suggest you present these two research questions in different manuscripts. We really appreciate your suggestion. We decided to present the two questions in the same manuscript because both issues are related, since vaccination coverage had a great influence on the decrease of COVID-19 cases. Therefore, we found it interesting to analyze which factors were associated with vaccination. We completely agree that there is a lot of information on display. Therefore, changes have been made to focus on these two main questions and to avoid too much additional information.

Reviewer #3: 

1. This study describes a large cohort of HIV+ patients over 2 years to determine incidence and risk factors for sars cov2 infection. Data taken from an established data base with informed consent. 96% patients remained in the cohort at the end of the study period (3411). The questions set out are investigated and answered well for the purpose of this manuscript. A few comments:

3. Table 1 : what is p value referring to? eg age; origin and transmission – P-value of table 1 refers to chi square in categorical variables and Mann-Whintey U test in continues. 

4. interesting that COPD is not a risk factor is there any data re Lung function in this cohort? All patients in our cohort with a diagnosis of COPD have respiratory function tests that support this diagnosis.

5. Are there any data re patients using HCV drugs? Thank you very much for your interest in our cohort. A total of 28 patients received HCV drugs during the study period. However, we did not analyze their association with SARS-CoV-2 infection.

4. There are other arv therapies which are thought to have sars-cov 2 activity. eg Abacavir and zidovudine and Protease inhibitors. Maybe include a comment or reference to this. Especially for those patients who were on 2nd / 3rd line regimens, and for patients with vl>50. Thank you for pointing this out. We have considered your suggestion and some new references have been included in the introduction. However, as we only analyzed the association between SARS-CoV-2 and tenofovir, we have not discussed other antiretroviral drugs in order not to overextend the manuscript.

Thank you again for your attention and consideration. We hope that the revised version of the manuscript can meet your expectations. Please let us know if further improvements are needed.

Best regards,

Francisca Artigues Serra & coauthors

---

## [Decision Letter · Decision Letter 1]

25 Jul 2024

Factors associated with SARS-CoV-2 infection among people living with HIV: Data from the Balearic cohort (EVHIA)

PONE-D-23-43118R1

Dear Dr. Artigues Serra,

We’re pleased to inform you that your manuscript has been judged scientifically suitable for publication and will be formally accepted for publication once it meets all outstanding technical requirements.

Kind regards,

Jake M. Pry, PhD, MPH

Academic Editor

PLOS ONE

Additional Editor Comments (optional):

Many thanks for taking time and making the effort to respond thoroughly to reviewer feedback.

Reviewers' comments:

Reviewer's Responses to Questions

**Comments to the Author**

1. If the authors have adequately addressed your comments raised in a previous round of review and you feel that this manuscript is now acceptable for publication, you may indicate that here to bypass the “Comments to the Author” section, enter your conflict of interest statement in the “Confidential to Editor” section, and submit your "Accept" recommendation.

Reviewer #2: All comments have been addressed

2. Is the manuscript technically sound, and do the data support the conclusions?

Reviewer #2: Yes

3. Has the statistical analysis been performed appropriately and rigorously? 

Reviewer #2: Yes

4. Have the authors made all data underlying the findings in their manuscript fully available?

Reviewer #2: Yes

5. Is the manuscript presented in an intelligible fashion and written in standard English?

Reviewer #2: Yes

6. Review Comments to the Author

Reviewer #2: (No Response)

7. PLOS authors have the option to publish the peer review history of their article (what does this mean?). If published, this will include your full peer review and any attached files.

Reviewer #2: No

---

## [Editor Report · Acceptance letter]

29 Jul 2024

PONE-D-23-43118R1 

PLOS ONE

Dear Dr. Artigues Serra, 

I'm pleased to inform you that your manuscript has been deemed suitable for publication in PLOS ONE. Congratulations! Your manuscript is now being handed over to our production team.

Kind regards, 

on behalf of

Dr. Jake M. Pry 

Academic Editor

PLOS ONE